# Foaming with Starch: Exploring Faba Bean *Aquafaba* as a Green Alternative

**DOI:** 10.3390/foods12183391

**Published:** 2023-09-11

**Authors:** Josseline S. Ramos-Figueroa, Timothy J. Tse, Jianheng Shen, Sarah K. Purdy, Jae Kyeom Kim, Young Jun Kim, Bok Kyung Han, Ji Youn Hong, Youn Young Shim, Martin J. T. Reaney

**Affiliations:** 1Department of Food and Bioproduct Sciences, University of Saskatchewan, Saskatoon, SK S7N 5A8, Canadamartin.reaney@usask.ca (M.J.T.R.); 2Department of Food and Biotechnology, Korea University, Sejong 30019, Republic of Korea; nutrigenomics@korea.ac.kr (J.K.K.); yk46@korea.ac.kr (Y.J.K.); hanmoo@korea.ac.kr (B.K.H.); khjy1025@korea.ac.kr (J.Y.H.)

**Keywords:** faba bean, *aquafaba*, chickpea, physicochemical property, vegan

## Abstract

The demand for sustainable and functional plant-based products is on the rise. Plant proteins and polysaccharides often provide emulsification and stabilization properties to food and food ingredients. Recently, chickpea cooking water, also known as *aquafaba*, has gained popularity as a substitute for egg whites in sauces, food foams, and baked goods due to its foaming and emulsifying capacities. This study presents a modified eco-friendly process to obtain process water from faba beans and isolate and characterize the foam-inducing components. The isolated material exhibits similar functional properties, such as foaming capacity, to *aquafaba* obtained by cooking pulses. To isolate the foam-inducing component, the faba bean process water was mixed with anhydrous ethanol, and a precipitated fraction was obtained. The precipitate was easily dissolved, and solutions prepared with the alcohol precipitate retained the foaming capacity of the original extract. Enzymatic treatment with α-amylase or protease resulted in reduced foaming capacity, indicating that both protein and carbohydrates contribute to the foaming capacity. The dried precipitate was found to be 23% protein (consisting of vicilin, α-legumin, and β-legumin) and 77% carbohydrate (amylose). Future investigations into the chemical structure of this foam-inducing agent can inform the development of foaming agents through synthetic or enzymatic routes. Overall, this study provides a potential alternative to *aquafaba* and highlights the importance of exploring plant-based sources for functional ingredients in the food industry.

## 1. Introduction

In the food industry, there is a growing preference for plant-based products that possess foaming, emulsification, and stabilization properties, as opposed to synthetic or animal-based alternatives. This shift is largely due to increased consumer demand for more sustainable plant-based ingredient options. To meet this demand, researchers have explored the functional properties of various biopolymers derived from sustainable sources, such as plant-based proteins, unmodified native starch [1], chemically modified starch [2,3], and polysaccharide-starch complexes [4]. These alternative ingredients have been tested for their ability to replicate the water-holding capacity and fat-binding properties of traditional animal-derived ingredients like eggs, meat, and milk [5,6].

The use of wastewater decanted from cooked chickpea, or *aquafaba*, as a foaming and emulsifying agent has gained popularity on the internet [7]. This process has been replicated by several food laboratories to produce a more consistent and standardized clean-label product [8]. Aquafaba is composed mainly of proteins and carbohydrates [7], but the specific types of proteins and carbohydrates have only been described qualitatively [7]. Therefore, further investigation is necessary to identify all constituents of *aquafaba* and to evaluate which legumes may possess comparable or superior functional properties, such as foaming capacity, stability, and emulsification, to those of *aquafaba* products [7].

Faba bean (*Vicia faba* L.) is the most widely grown legume after peas, chickpeas, and lentils [9]. It is a versatile crop that is largely grown in Mediterranean-type climates, including cold-temperate regions [10]. It is a significant food source in several developing countries in the Mediterranean region and parts of Latin America [11,12] and is also used in the production of animal feed in industrialized countries [13]. However, the functional properties of the faba bean can be affected by genotype and environmental variables, including temperature and water deficit [14]. Faba bean is a highly nutritious pulse, rich in choline, lecithin, minerals (e.g., calcium, magnesium, iron, copper, zinc, manganese), and secondary metabolites (e.g., phenolic compounds) [15,16,17]. Faba bean seeds are primarily composed of carbohydrates (~40–60%), protein (~30%) [9,18], and minerals and fat (~2%) [19]. The carbohydrate fraction is comprised of 20–40% starch [9], with faba bean starch exhibiting higher amylose and resistant starch content than other legumes [9]. However, dehulling and soaking of faba bean can reduce anti-nutritional compounds, including enzyme inhibitors, which could affect the functional properties of faba bean [20]. In addition, faba bean protein has been investigated for its emulsification properties in food production [21]. Interestingly, *aquafaba* production from peas [22] and lima beans [23] has been investigated; although, to our knowledge, *aquafaba* from faba beans has neither been investigated nor characterized. Due to the similar seed composition of faba bean to chickpea seed [24], faba bean could be a valuable pulse for use in food formulations [25], such as a foaming, emulsifying, or thickening agent due to its unique properties, e.g., high gelation temperature, fast retrogradation, and gel elasticity.

The focus of this study was to characterize the different fractions obtained from faba bean process water and to investigate its functional properties compared to sustainable *aquafaba* (i.e., chickpea wastewater). We analyzed this solution for its functional properties, such as foaming capacity and stability, characterized its composition in terms of starch, monosaccharide, protein, and peptide content, and evaluated the molecular weight (MW) of the carbohydrates, starch–protein complexes, and particle size distributions. By characterizing faba bean process water in detail, we aim to facilitate its use in the formulation of novel food products, including as a vegan-friendly alternative.

## 2. Materials and Methods

### 2.1. Materials and Chemicals

The dry faba bean (*Vicia faba* L.) seeds used in this study were provided by Prairie Tide Diversified Inc. (Saskatoon, SK, Canada). Anhydrous ethanol was purchased from Greenfield Global Inc. (Brampton, ON, Canada). For experimentation, a solution of industrial α-amylase and glucoamylase was generously provided by Federated Co-operatives Ltd. (formerly Terra Grain Fuels Inc., Belle Plaine, SK, Canada). Additionally, high-purity α-amylase was also obtained from Alfa Aesar (Tewksbury, MA, USA) and Sigma-Aldrich (Oakville, ON, Canada) for use as analytical reference standards. The chemicals used in this study were purchased from Sigma-Aldrich (Oakville, ON, Canada) and include D-glucose, D-mannose, D-xylose, D-fucose, D-galactose, D-arabinose, L-rhamnose, *myo*-inositol, sodium borohydride (NaBH_4_), trifluoro acetic acid (TFA), acetic anhydride (Ac_2_O), pyridine, glacial acetic acid, 1-methylimidazol, sodium sulfate (Na_2_SO_4_), protease (from *Aspergillus oryzae*), 2,5-dihydroxybenzoic acid (DHB), and pullulan standards of known molecular weights (1.39 and 200 kDa).

For nuclear magnetic resonance (NMR) analyses, deuterium oxide (D_2_O), deuterated dimethylsulfoxide (DMSO-d_6_), and 3-(trimethylsilyl)propionic-2,2,3,3-d_4_ acid sodium salt (TSPA) were purchased from Sigma-Aldrich (Oakville, ON, Canada). The Bradford protein assay dye reagent and sodium dodecyl sulphate polyacrylamide gel electrophoresis (SDS-PAGE) buffers were purchased from Bio-Rad Laboratories (Hercules, CA, USA). Finally, the PageRuler™ prestained protein ladder (10–180 kDa) was purchased from Thermo Fisher Scientific (Waltham, MA, USA).

### 2.2. Faba Bean Dehulling

Dried beans were initially milled to separate the hull from the kernel. Subsequently, the dehulled faba bean kernel and the skin were air classified, using a custom-built air classifier. The process involved placing a mixture of hulls and kernels into a 3-inch-wide polyvinyl chloride (PVC) tube and allowing it to fall against an air current. The air velocity was adjusted to ensure that the cotyledons settled at the bottom of the classifier, while the hulls were blown through an inverted U-bend into a collector at the top. Once separated, the cotyledons were stored at room temperature for further processing.

### 2.3. Aqueous Extraction of Dehulled Faba Bean Cotyledons

A simple and efficient aqueous extraction method was developed to isolate various components from the dehulled faba bean cotyledons. Briefly, 100 g of milled split faba bean cotyledons were mixed with 200 mL of water and heated on a stovetop under constant stirring until boiling for approximately 45 min, or until the liquid was reduced to approximately 100 mL. The resulting mixture of faba bean debris and liquid was then filtered through a kitchen strainer (pore size of 1 mm) and cooled to room temperature. Next, the filtrate was subjected to centrifugation at 4500 rpm for 45 min at 4 °C, using a Labnet Spectrafuge 24D Digital Microcentrifuge (Labnet International Inc., Edison, NJ, USA). The supernatant (faba bean *aquafaba*; FBA), was carefully decanted into a separate container and stored at 4 °C until being further processed.

### 2.4. Isolation of Starch and Protein from FBA through Ethanol Treatment

These methods were developed based on our finding that FBA foaming properties decreased with the addition of α-amylase. To isolate the starch and protein components of FBA and investigate their contribution to its foaming properties, FBA (15 mL) was mixed with food-grade anhydrous ethanol (45 mL). Food-grade anhydrous ethanol was used as it is considered food-safe, unlike other organic solvents. The suspension was then centrifuged at 4500 rpm for 45 min at 4 °C and the FBA supernatant (FBA-S) was collected. The excess ethanol in the FBA-S was removed via rotary evaporation, and the resulting solution was divided into two aliquots. One aliquot was used to assess the foaming capacity and stability, while the other was freeze-dried and stored in a desiccator for instrumental analysis. Meanwhile, an additional 10 mL of anhydrous ethanol was added to the remaining pellet (solid fraction) to remove excess ethanol-soluble components. The sample was centrifuged again under the same conditions, and the supernatant was discarded. The resulting pellet was resuspended in water and dried via lyophilization using a FreezeZone 12 freeze-dryer (Labconco, Kansas City, MO, USA). The dried precipitate fraction of FBA (FBA-P) was then stored in a desiccator at room temperature until use. Similarly, FBA-P was dissolved in 1% (*w*/*v*) water and assayed for its foaming capacity and stability. Both freeze-dried fractions (FBA-P and FBA-S) were then characterized as described in the following sections.

### 2.5. Foam Capacity and Stability

The foaming capacity (FC) and foaming stability (FS) were assessed similarly as described in Mustafa et al. 2018 [26]. Briefly, faba bean process water (5 mL) was diluted with 10 mL of water and whipped for 2 min using a Kitchen Aid Ultra Power Mixer (Kitchen Aid, St. Joseph, MI, USA) with a speed setting of 10. The foam volume was then measured immediately after whipping (*V*_F0_; at time = 0 min) and after 30 min (*V*_F30_) using a graduated cylinder. FC and FS were calculated using Equations (1) and (2), respectively.
(1)$$\%FC=\frac{V_{F0}}{V_{sample}}\times 100$$
(2)$$\%FS=\frac{V_{F30}}{V_{F0}}\times 100$$

### 2.6. FBA-P and Starch Standard ^1^H-NMR Analysis

To identify the compounds present in FBA-P, approximately 1 mg of FBA-P was dissolved in D_2_O and separately in a mixture of DMSO-d_6_ and D_2_O. The ^1^H-NMR spectra were recorded using a Bruker Advance III HD 600 MHz spectrometer (NMR Bruker, Mississauga, ON, Canada) and analyzed using the TopSpin™ 3.2 software (Bruker BioSpin GmbH, Billerica, MA, USA). A solution of high-purity α-amylose was used as a standard. Lyophilized FBA-S was also dissolved in D_2_O and subjected to ^1^H-NMR analysis to identify compounds present in these fractions.

### 2.7. Analysis of Monosaccharide Content by α-Amylase Hydrolysis and ^1^H-NMR Analysis

To determine the monosaccharide content in FBA-P, 0.1 g of freeze-dried sample was dissolved in 10 mL of reagent-grade water (Optima LC/MS water), and a drop of industrial α-amylase or glucoamylase concentrate was added. The mixture was vortexed and incubated in a 60 °C water bath for 3 h and cooled to room temperature prior to centrifugation at 4500 rpm for 40 min at 19 °C. The supernatant was collected, frozen, and lyophilized prior to storage in a desiccator at room temperature, while the pellet was discarded. An aliquot of the supernatant was mixed with D_2_O prior to ^1^H-NMR analysis to determine the monosaccharide content. Another aliquot of the lyophilized sample was separately prepared for gas chromatography equipped with a flame ionization detector (GC-FID), described below).

### 2.8. Monosaccharide Content Determined by Gas Chromatography

Monosaccharide content was further qualified using GC-FID. Standard stock solutions (50 mg/mL) of D-glucose, D-mannose, D-xylose, D-fucose, D-galactose, D-arabinose, L-rhamnose, and *myo*-inositol (reference standard) were prepared using reagent-grade water. An aliquot (20 µL) of each stock solution was transferred to a 2 mL glass vial and combined with 50 µL of a freshly prepared 1% NaBH_4_ (*w*/*v*) solution. The mixture was incubated at 50 °C for 1.5 h and shaken every 30 min. After incubation, the sample was cooled to room temperature, and glacial acetic acid was added dropwise to the mixture to quench excess NaBH_4_. The samples were then acetylated by adding Ac_2_O (50 µL) and 1-methylimidazole (10 µL), which was shaken and incubated at 45 °C for 30 min. Excess Ac_2_O was quenched by adding water (700 µL), and the acetylated product was extracted by adding 600 µL of dichloromethane (DCM) and vortexing for 20 s. The mixture was then incubated at room temperature to permit the separation of water and DCM fractions. The water layer was removed, and anhydrous Na_2_SO_4_ was added to the DCM fraction to remove excess water. Finally, the solution was filtered through a 13 mm (0.2 µm) PTFE filter prior to analyses via GC-FID.

To analyze the monosaccharide content of FBA-P (starch samples), 10 mg of the freeze-dried sample was hydrolyzed using 1 mL of 4 M TFA and incubated at 100 °C for 4 h in oven-dried Chemglass pressure refill reaction vials (VWR, Edmonton, AB, Canada) to prevent pressure build-up and sample evaporation. After incubation, the samples were cooled to room temperature, and a 100 µL hydrolyzed sample was mixed with 100 µL of methanol and evaporated at 100 °C for 30 min. This was repeated 3 times before the dried residue was dissolved in 20 mL of DMSO and 20 mL of water. To reduce any sugars obtained during hydrolysis, 80 mL of freshly prepared 1% NaBH_4_ (*w*/*v*) was added to the sample, and the mixture was incubated at 50 °C for 1.5 h. The sample was then filtered through a 13 mm (0.2 mm) PTFE filter prior to GC-FID analyses.

### 2.9. GC-FID Analytical Methodology

Monosaccharide analyses were conducted using an Agilent 6850 GC (Santa Clara, CA, USA) equipped with a flame ionization detector (FID). Sample injection was carried out using an Agilent 6850 (Santa Clara, CA, USA) automatic liquid sampler with a 10 µL Hamilton cemented needle syringe (Reno, NV, USA). The separation of monosaccharides (Table 1) was achieved on a 30-m Agilent J&W DB-5 fused-silica column (ID 0.25 mm, 0.25 µm film). The FID was operated at 300 °C with flow rates of 136.0 mL min^−1^ of air, 35 mL min^−1^ of hydrogen, and 45 mL min^−1^ nitrogen. The initial oven temperature was 60 °C and was maintained for 2 min. The temperature was increased to 150 °C at 20 °C min^−1^ and then further increased to 300 °C at 6 °C min^−1^ and maintained for 3.5 min, giving a total run-time of 35 min. Helium was used as the carrier gas, with a flow rate of 1.2 mL min^−1^. The inlet was operated in splitless mode at 290 °C, and 1 µL of the sample was injected. The syringe was thoroughly washed with hexane between injections to avoid cross-contamination.

### 2.10. Protein Content of FBA

Protein content in FBA-P was determined and corroborated using three methods: (1) ^1^H-NMR analyses; (2) inference via total nitrogen content; and (3) the Bradford protein assay. SDS-PAGE was then performed to determine the approximate size of the protein components in FBA-P.

#### 2.10.1. Peptide Analysis via ^1^H-NMR Analysis

Freeze-dried FBA-P and FBA-S samples were dissolved in D_2_O and analyzed by ^1^H-NMR using a 600 MHz NMR spectrometer described above.

#### 2.10.2. Determination of Total Nitrogen Content Using LECO Combustion Analyzer

To determine the total nitrogen content of freeze-dried FBA-P, an automated LECO nitrogen combustion analyzer (FP-528, LECO, St. Joseph, MI, USA) was used. The analyzer was equipped with an induction furnace and a thermal conductivity detector. The calibration standard used was ethylenediaminetetraacetic acid (EDTA) (LECO), and the carrier gases were helium (He), compressed air, and oxygen (O_2_) with a purity of 99.99% (Linde, Saskatoon, SK, Canada). The sample was loaded into the loading head and purged with atmospheric gases, while the ballast volume and gas lines were also purged. The sample was then dropped into the hot furnace (850–900 °C) and rapidly combusted with pure O_2_. The products of combustion were CO_2_, H_2_O, NO_x_, and N_2_. A portion of the combustion gases was passed through a copper catalyst to remove oxygen, scrub moisture and CO_2_, and convert nitrous oxides to N_2_. The nitrogen content was determined by thermal conductivity. The protein content in FBA-P was estimated by multiplying the nitrogen content by a conversion factor of 6.25 [27].

#### 2.10.3. Total Protein Content via Bradford Protein Assay

A standard curve was generated using the Bio-Rad protein assay dye reagent concentrate (200 µL) mixed with 1 mL of standard solutions containing 0, 2, 4, 6, 8, and 10 µg of bovine serum albumin. The absorbance of each solution was recorded at 595 nm using a UV-Vis spectrophotometer. The same procedure was repeated for 1 mL of FBA-P solution containing 1 µg of FBA-P in water.

### 2.11. Separation of Protein via SDS-PAGE

To determine the approximate size of protein and components present in FBA-P, SDS-PAGE was performed using a 15% resolving gel and a 5% stacking gel. Freshly prepared 10% ammonium persulfate and TEMED were used to prepare the gels. A 10 mg/mL solution of FBA-P in water (15 mL) was mixed with 5 µL of the marker and incubated at 95 °C for 10 min. The running conditions started at 80 V for 20 min, prior to increasing to 100 V for 1 h. Once the run was complete, the gel was stained with a 0.2% (*w*/*v*) solution of brilliant blue in a 4:1 mixture of methanol and acetic acid and destained in a pure solution of 4:1 mixture of methanol and acetic acid.

### 2.12. Molecular Weight Analysis of FBA-P via DOSY ^1^H-NMR

To determine the MW of FBA-P, we utilized diffusion-ordered spectroscopy (DOSY) ^1^H-NMR analysis. Approximately 1 mg of the sample was dissolved into 1 mL of DMSO-d_6_. The DOSY experiments were performed at 300 K on a Bruker AVANCE III AQS600 NMR spectrometer operating at 600 MHz and equipped with a Bruker multinuclear *z*-gradient inverse probe head capable of producing gradients in the *z* direction with a strength of 55 G/cm. DOSY experiments were performed, similarly as described by Kleks et al. 2021 [28], using a stimulated echo sequence incorporating bipolar gradient pulses and a longitudinal eddy current delay (BPP-LED). The gradient strength was logarithmically incremented in 16 steps from 2% up to 98% of the maximum gradient strength. Diffusion times and gradient pulse durations were optimized for each experiment to achieve a 95% decrease in the resonance intensity at the largest gradient amplitude; typically, diffusion times between 150 and 200 ms and bipolar rectangular gradient pulses between 1.5 and 2.3 ms were employed. After Fourier transformation and baseline correction, the diffusion dimension of the 2D-DOSY spectra was processed using the Bruker Dynamics Center (version 2.7). In this experiment, the diffusion rates of the analytes in the solution were directly correlated to their molecular weights. Pullulan standards (MW of 1.39 and 200 kDa) were used to evaluate the FBA-P material.

### 2.13. Particle Size Distribution of FBA-P

Particle-size distribution of freeze-dried FBA-P samples was measured using a Mastersizer 2000 laser static light scattering instrument (Malvern Instruments Ltd., Malvern, UK) equipped with a Hydro 2000S system for handling samples. FBA-P (1.47 g) was suspended in 100 mL of water and stirred for 5 min prior to analysis.

### 2.14. Protein–Starch Complex Interdependence for Foaming Ability

A suspension of 0.10 to 0.14 g of FBA-P in 5 mL of water was treated with 50 µL of protease from *A. oryzae* (600 U/mL) and incubated for 30 min. After this time, the bubbles formed decreased dramatically. The resulting mixture was submitted for foaming capacity analysis and centrifuged. The supernatant was recovered and freeze-dried prior to ^1^H-NMR analyses.

## 3. Results and Discussion

### 3.1. Aquafaba Production, Functional Precipitates, and Starch Contain

Foam with functional properties like *aquafaba* can be generated from the process water obtained during the mild cooking of milled faba beans (FBA). Although previous studies have investigated the composition of similar solutions, such as *aquafaba*, the compound responsible for inducing foam in these solutions has remained unknown. In this study, we aimed to identify the foam-inducing compound in FBA. Faba bean process water was treated with ethanol to obtain precipitate (FBA-P) and supernatant (FBA-S) fractions. Notably, only the FBA-P fraction retained its foaming properties, suggesting that the foam-inducing compound may be present in this fraction.

Dehulled faba bean kernels were soaked overnight and exhibited a ~2-fold increase in mass. Upon cooking the soaked kernels for 45 min, an extract of 375 mL of FBA was obtained from 570 g of cooked faba bean. Treatment of the extract with ethanol generated 0.95 g of FBA-P and 193 mL of FBA-S per 100 mL of FBA. Upon lyophilization, the dried FBA-S resulted in 6.48 g of powder per mL of FBA. The foaming capacities and stabilities of the aqueous extract, precipitate, and supernatant after ethanol treatment are presented in Table 2. Interestingly, faba bean process water depleted in starch demonstrated a significant reduction.

We found two ^1^H-NMR spectra for the α-amylose standard and FBA-P (Figure 1). FBA-P and an α-amylose reference standard were dissolved in DMSO-d_6_ prior to analysis by ^1^H-NMR. The resulting ^1^H-NMR spectra supported the presence of starch as the major DMSO-soluble component of FBA-P (Figure 1). Therefore, further investigations were conducted to explore the physicochemical properties of both the starch and protein components of FBA-P.

The NMR spectrum of the α-amylose reference standard and FBA-P exhibited similarities, suggesting that the sample was primarily composed of amylose units. To confirm whether FBA-P consisted of glucose units, enzymatic hydrolysis was performed using α-amylose and glucoamylase. The former enzyme can cleave α-1,4-glycosidic bonds, while the latter can hydrolyze α-1,6-glycosidic bonds and α-1,4-glycosidic bonds. α-amylase treatment of FBA-P generated free glucose and maltose (Figure 2a), whereas glucoamylase cleaved only glucose units (Figure 2b). Interestingly, α-1,6-disaccharide products (e.g., isomaltose and gentiobiose), which are expected for branched starches, were not observed, suggesting that the starch component of FBA-P was mainly composed of glucose units via α-1,4-glycosidic bonds.

### 3.2. Monosaccharide Content Determined by Gas Chromatography

To further characterize the monosaccharide units in the starch component of FBA-P, acid hydrolysis, using TFA, was conducted. Treatment of FBA-P with TFA at high temperatures did not completely solubilize the sample, indicating the presence of non-hydrolyzable components (e.g., proteins). Therefore, to determine the monosaccharide profile of FBA-P (Table 1), the mixture was filtered, dried, derivatized, and acetylated prior to GC-FID analyses. The results confirmed that the starch component of FBA-P primarily consisted of glucose units, further supporting the presence of amylose in FBA-P.

### 3.3. Separation of Protein via SDS-PAGE

The main carbohydrate component of FBA-P was determined to be amylose, representing ~77%, when hydrolyzed with α-amylase and glucoamylase. In addition, GC-FID analysis confirmed that the main monomeric carbohydrate component of FBA-P was glucose. The total protein content in FBA-P was estimated to be between 22 and 25% (*w*/*w*) based on total N content and the Bradford protein assay. SDS-PAGE identified the protein profile of FBA-P, revealing a mixture of proteins with molecular weights primarily of approximately 45, 36, and 27 kDa, along with minor proteins ranging from 15 to 25 kDa (Figure 3).

Meanwhile, after ethanol treatment, FBA-S retained small molecules that were solubilized (see Section 3.4).

### 3.4. Molecular Weight Analysis of FBA-P via DOSY ^1^H-NMR

The MW of the soluble amylose in FBA-P was determined using DOSY ^1^H-NMR, which revealed an MW of 10.3 kDa (Figure 4). Commercial pullulan standards with molecular weights of 1.39 and 200 kDa were used as reference standards. Particle size distribution was analyzed using dynamic light scattering, which showed two distinct populations: One with particles of approximately 10 µm and the other with particles of ~100 µm (Figure 5). The refractive indices of the particles and dispersant (water) were 1.53 and 1.33, respectively. The percent obscurity was 12.55%, and the volume-weighted mean particle size (D [3,4]) was measured in triplicate. 

The protein profile of FBA-P was characterized to primarily consist of vicilin, α-legumin, and β-legumin [29]. Our investigation revealed that both protein and starch components were necessary for the foaming properties observed for FBA-P. Although further confirmatory investigations should be conducted, we hypothesized three mechanisms that are required to be inhibited to achieve good foaming properties: (1) Disproportionation of bubbles, (2) coalescence of bubbles due to film instability, and (3) water drainage from the surface of the bubbles leading to protein removal from the bubble’s film [30]. The presence of the soluble low MW amylose can reduce foam collapse by preventing water drainage from the surface of the bubbles. Amylose forms coiled structures [31] that are hydrophilic on the surface [32], thus increasing its water retention properties.

Lastly, to evaluate the effects of the protein–starch interactions on the foaming properties of the mixture, FBA-P was treated with protease; however, no foaming was observed in the mixture, suggesting the presence of protein was essential for the observed foaming properties in FBA-P. In addition, ^1^H-HMR analyses revealed the presence of amino acids, while the precipitate was primarily composed of polysaccharides. Similarly, enzymatic hydrolysis of FBA-P using α-amylase resulted in the elimination of its foaming properties.

### 3.5. FBA-S Analysis

Foaming capacity analysis revealed that FBA-S had no foaming potential (Table 2). Additionally, ^1^H-NMR analysis of FBA-S identified the presence of residual ethanol along with significant amounts of various organic compounds, including sucrose, raffinose, stachyose and verbascose, vicine, convicine, citric acid, and levodopa (*L*-Dopa) (Table 3, Figure 6).

The presence of *L*-Dopa is of interest, as this compound is a dopamine precursor, which has been used in the treatment of Parkinson’s disease and hormonal imbalance [17,33,34,35,36,37,38]. The synthesis of this compound can be influenced by soil nitrogen and phosphorus content during growth [39] and is also prone to degradation during cooking or in the presence of alkaline solutions [40,41,42]. Nonetheless, isolating and enriching this value-added compound from faba bean process water can provide producers with new market entries.

## 4. Conclusions

Faba bean process water was characterized to assess its foaming capacity and composition. Ethanol extraction of FBA yielded two fractions, FBA-P and FBA-S. FBA-P, mainly composed of starch (predominantly low MW amylose; 77%) and minor proteins (23%), exhibited significant foaming properties, indicating that the foaming capacity of FBA-P was dependent on the formation of starch–protein complexes. However, upon hydrolysis of proteins using protease, the foaming capacity of FBA-P was significantly reduced. In contrast, FBA-S did not demonstrate foaming capabilities due to a lack of starches and proteins. However, FBA-S could be used as a source of *L*-Dopa for medicinal applications. Altogether, faba bean process water demonstrated similar functional properties as sustainable *aquafaba* (chickpea wastewater), offering a vegan-friendly alternative ingredient in the formulation of food products. The presence of *L*-Dopa also suggests further investigations of this material as an added-value product for its medicinal applications, such as in the treatment of Parkinson’s disease.

## Figures and Tables

**Figure 1 foods-12-03391-f001:**
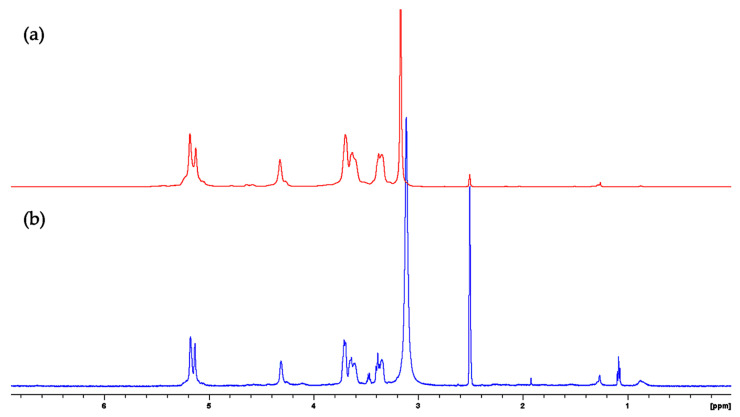
^1^H-NMR spectra of (**a**) α-amylose standard and (**b**) FBA-P dissolved in DMSO-d_6_.

**Figure 2 foods-12-03391-f002:**
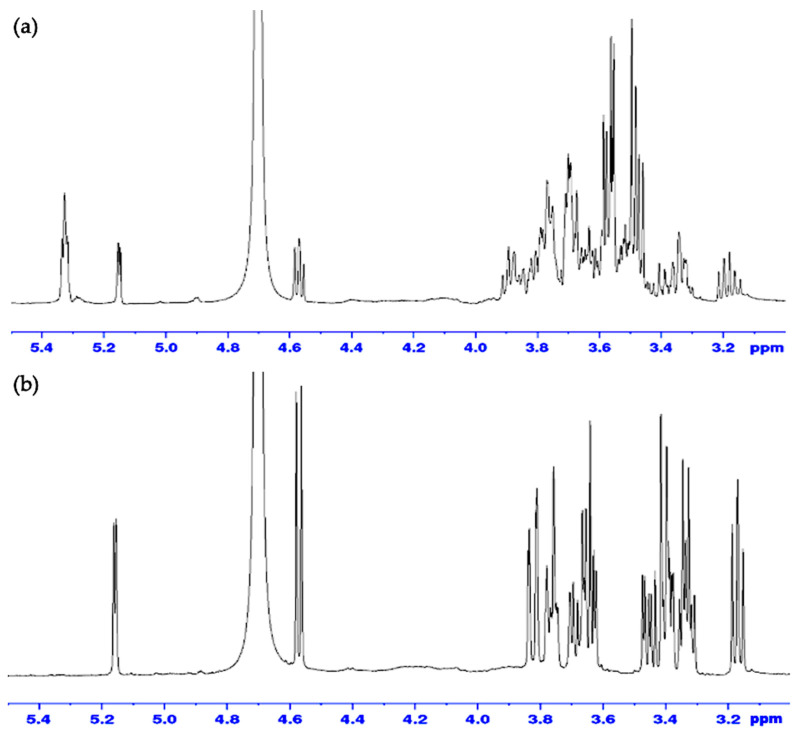
H^1^-NMR spectra of FBA-P treated with (**a**) α-amylase and (**b**) glucoamylase.

**Figure 3 foods-12-03391-f003:**
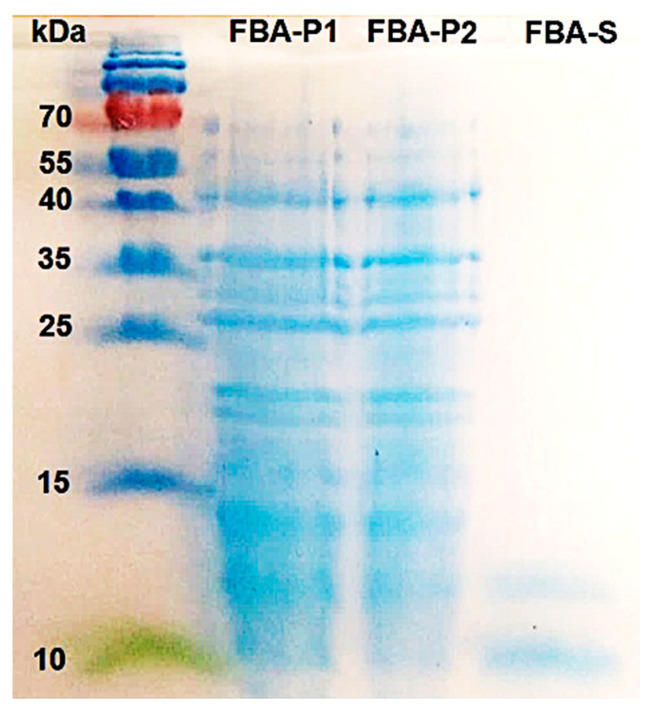
Protein profile of FBA-P and FBA-S as determined by SDS-PAGE. Numbers on the left side of the gel indicate the position and mass of the molecular weight marker, in kDa. The FBA-P fraction was submitted in duplicates.

**Figure 4 foods-12-03391-f004:**
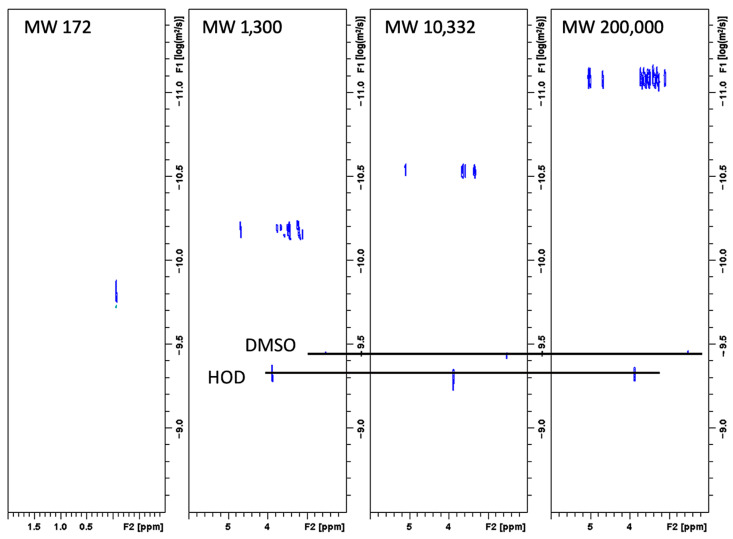
2D DOSY spectra recorded in DMSO at 298 K of TSPA (MW = 172 g/mol), pullulan standard-53,168 (MW = 1300 g/mol), pullulan standard-01615 (MW = 200,000 g/mol), and the FBA-P (calculated MW = 10,332 g/mol). TSPA: 3-(trimethylsilyl)propionic-2,2,3,3-d_4_ acid sodium salt.

**Figure 5 foods-12-03391-f005:**
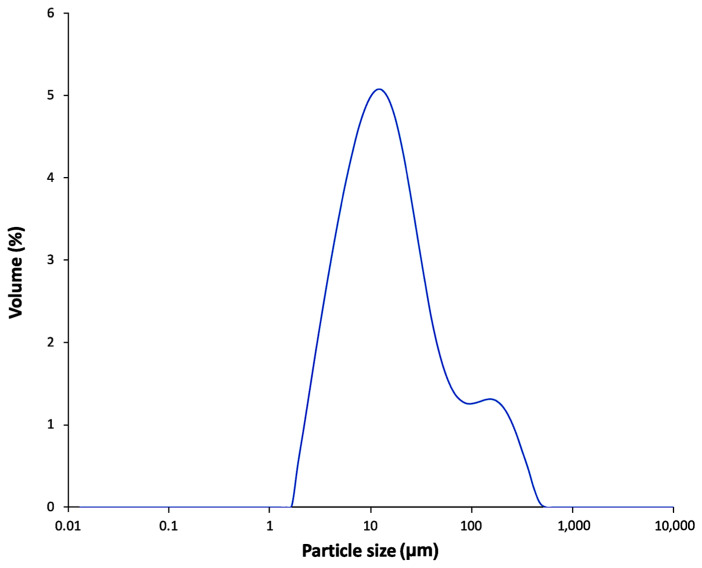
Particle size of a solution of FBA-P in water (1 mg/mL).

**Figure 6 foods-12-03391-f006:**
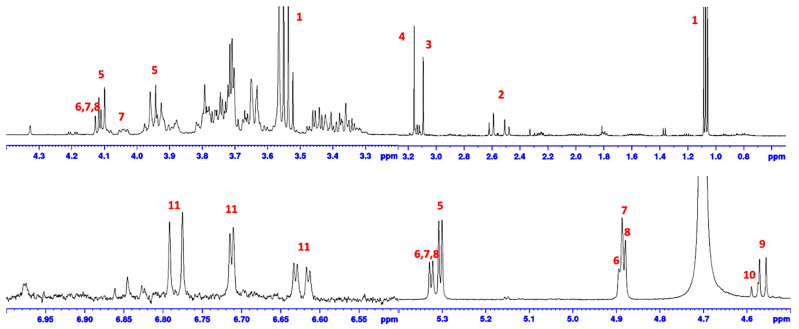
^1^H-NMR diagnostic resonances of compounds present in FBA-S. 1, ethanol; 2, citric acid; 3, choline; 4, betaine; 5, sucrose; 6 raffinose; 7, stachyose; 8, verbascose; 9, vicine; 10, convicine; 11, *L*-Dopa.

**Table 1 foods-12-03391-t001:** Retention times of monosaccharide standards.

Compound	Retention Time (min)
D-arabinose	14.286
D-fucose	14.212
D-galactose	17.869
D-glucose	17.657
D-mannose	17.606
L-rhamnose	14.053
D-xylose	14.503
TFA hydrolyzed FBA-P	17.666

**Table 2 foods-12-03391-t002:** Yield of FBA, FBA-P, FBA-S, and their foaming characteristics.

Characteristics	Yield (%)	Foaming Capacity (%)	Foaming Stability (%)
FBA	87 ^a^	600.01 ± 54.98	94.44 ± 8.10
FBA-P	0.62 ^b^ 0.95 ^c^	533.33 ± 43.32	93.75 ± 7.12
FBA-S	1.73 ^d^ 1.14 ^e^	NA	NA

^a^ Volume of aqueous extract (FBA) per 100 g of soaked kernel. ^b^ Mass of precipitate after ethanol treatment (FBA-P) per 100 g of soaked kernel. ^c^ Mass of FBA-P per 100 mL of FBA. ^d^ Mass of dry supernatants after ethanol treatment (FBA-S) per 100 mL of FBA. ^e^ Mass of dry FBA-S per 100 g of soaked kernel. NA: Not applicable. Values are mean ± standard deviation (SD, *n* = 3).

**Table 3 foods-12-03391-t003:** Composition of FBA-S.

Compound	Profile (%)
Ethanol	7.61
Citric acid	5.31
Phosphocholine	2.12
Betaine phosphate	3.39
Sucrose	24.41
Raffinose	7.63
Stachyose	9.98
Verbascose	24.82
Vicine	9.75
Convicine	3.70
*L*-Dopa	1.30

## Data Availability

The data from the current study are available from the corresponding author upon reasonable request.

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
