# Peer review of "Foaming with Starch: Exploring Faba Bean Aquafaba as a Green Alternative"

_foods, 2023, doi:10.3390/foods12183391_

Round 1
Reviewer 1 Report
Dear authors,
Please consider the comments for your review:
L50-71 - It is a long statement just to explain about fava bean, instead of highlighting the recent works that studied aquafaba from fava bean or other pulses.
L115 - Replace the ''?'' to the pore size
l116 - include brand, model, city and country of centrifuge used.
Topic 2.4 - Explain why did the authors use ethanol (to select protein in detriment of starch?)
L134-135 - Which findings? Where are the references?
L199 - This ''3. Results and Discussion'' was not supposed to be here.
Table 1 - The authors can transfer this table to R&D part, and add the columns for monosaccharides quantified in samples
L268 - correct the typo for aquafaba
Table 2/ L 289 - There is no ''ND'' in the Table 2.
Table 3 - Concentration is never expressed as percentage. The authors can express as mass per volume of product in here.
L350-351 - Explain the reason why of these 2 populations in particle size. Also, why FBA-S is not in figure 5?
L363-367 - In which figure/table the authors are supported?
L372 - If authors do not have data, there is not need to include the information.
L373 - In which figure/table the authors observed no foaming was observed?
L376 - alpha amylase; please correct typo
L391 - What do you mean when you refer to ''isolation''? I was not able to see a single molecule isolated. The authors obtained different fractions with multiple composition.
Additional comment: That would be interesting is authors include extraction of chickpea and analyze the aquafaba as a control. How comparable is fava bean with chickpea?
Author Response
Reviewer #1:
Comment: L50-71 - It is a long statement just to explain about fava bean, instead of highlighting the recent works that studied aquafaba from fava bean or other pulses.
Response: This paragraph has been revised to reduce general faba bean information and includes more recent citations. Lines 48-68 now reads:
“Faba bean (Vicia faba L.) is the most widely grown legume after pea, chickpea, and lentil [9]. It is a versatile crop that is largely grown in Mediterranean-type climates, including cold-temperate regions [10]. It is a significant food source in several developing countries in the Mediterranean region and parts of Latin America [11,12], as well as used in the production of animal feed in industrialized countries [13]. However, the functional properties of faba bean can be affected by genotype and environmental variables, including temperature and water deficit [14]. Faba bean is a highly nutritious pulse, rich in choline, lecithin, minerals (e.g., calcium, magnesium, iron, copper, zinc, manganese), and secondary metabolites (e.g., phenolic compounds) [15–17]. Faba bean seeds are primarily composed of carbohydrates (~40–60%), protein (~30%) [9,18], and minerals and fat (~2%) [19]. The carbohydrate fraction is comprised of 20–40% starch [9], with faba bean starch exhibiting higher amylose and resistant starch content than other legumes [9]. However, dehulling and soaking of faba bean, can reduce anti-nutritional compounds, including enzyme inhibitors, which could affect the functional properties of faba bean [20]. In addition, faba bean protein has been investigated for its emulsification properties in food production [21]. Interestingly, aquafaba production from peas [22] and lima beans [23] have been investigated; although, to our knowledge, aquafaba from faba beans has neither been investigated nor characterized. Due to the similar seed composition of faba bean to chickpea seed [24], faba bean could be a valuable pulse for use in food formulations [25]; such as a foaming, emulsifying, or thickening agent due to its unique properties (e.g., high gelation temperature, fast retrogradation, and gel elasticity.”
Comment: L115 - Replace the ''?'' to the pore size
Response: Pore size was 1 mm.
Comment: L116 - include brand, model, city and country of centrifuge used.
Response: Centrifuge information has been added. Line know reads “Next, the filtrate was subjected to centrifugation at 4500 rpm for 45 min 4 °C, using a Labnet Spectrafuge 24D Digital Microcentrifuge (Labnet International Inc., Edison, NJ, USA).”
Comment: Topic 2.4 - Explain why did the authors use ethanol (to select protein in detriment of starch?)
Response: Food grade anhydrous ethanol was used, as it is food safe, unlike other organic solvents. Lines 119-122 now reads “To isolate the starch and protein components of FBA and investigate their contribution to its foaming properties, FBA (15 mL) was mixed with food-grade anhydrous ethanol (45 mL). Food-grade anhydrous ethanol was used, as it is considered food-safe, unlike other organic solvents.”
Comment: L134-135 - Which findings? Where are the references?
Response: The findings are based on our own preliminary findings. Lines 135-137 have been clarified and now reads “These methods were developed based on our previous preliminary findings (Reaney, University of Saskatchewan, 2022, personal communication)…”
Comment: L199 - This ''3. Results and Discussion'' was not supposed to be here.
Response: The authors are confused with this comment. There is no “3. Results and Discussion” on Line 202.
Comment: Table 1 - The authors can transfer this table to R&D part, and add the columns for monosaccharides quantified in samples
Response: GC-FID was not used to quantify the different monosaccharide contents, but instead to confirm/qualify that the faba bean precipitate fraction consisted of monosaccharide units. Therefore, the authors believe that Table 1 should remain where it is, as it indicates the retention times for the GC-FID method.
Comment: L268 - correct the typo for aquafaba
Response: There is no typo for aquafaba. Throughout the manuscript aquafaba has been italicized to highlight this term.
Comment: Table 2. L 289 - There is no ''ND'' in the Table 2.
Response: ND has been switched to NA for Not Applicable.
Comment: Table 3 - Concentration is never expressed as percentage. The authors can express as mass per volume of product in here.
Response: The word “Concentration” in Table 3 has been replaced with “Profile (%)”.
Comment: L350-351 - Explain the reason why of these 2 populations in particle size. Also, why FBA-S is not in figure 5?
Response: Two Populations of Starch: The distinct presence of two starch populations, as revealed by the DOSY NMR analysis, can be attributed to variations in molecular size. Starch molecules are composed of amylose and amylopectin components that create polymeric structures. These components exhibit varying degrees of branching, molecular weight, and conformation, leading to fluctuations in their hydrodynamic radius. These variations influence their diffusion rates in the NMR experiment, resulting in discrete diffusion coefficients. As such, the two populations observed correspond to starch molecules with differing degrees of polymerization or structural arrangements, generating distinguishable peaks in the DOSY NMR spectrum.
Exclusion of FBA-S from Figure 5: Thank you for highlighting the absence of FBA-S in Figure 5. We acknowledge that this omission was unintentional and regret any confusion it may have caused. We would like to clarify the rationale behind this decision based on our initial response:
Commercial pullulan standards were deliberately selected for the molecular weight calibration, encompassing 1.39 and 200 kDa, thereby facilitating the identification of a broad spectrum of saccharide components. FBA-S, however, was not included in Figure 5 due to its classification as the supernatant fraction. Furthermore, FBA-S did not demonstrate any significant foaming capacity or foaming stability, as corroborated by the data presented in Table 2. Consequently, our focus was directed solely toward the analysis of the FBA-P (precipitate) fraction, given its more pronounced functional attributes and potential relevance.
Comment: L363-367 - In which figure/table the authors are supported?
Response: Clarification has been added to Lines 366-370 which now reads “Although further confirmatory investigations should be conducted, we hypothesized three mechanisms that are required to be inhibited to achieve good foaming properties: (1) disproportionation of bubbles, (2) coalescence of bubbles due to film instability, and (3) water drainage from the surface of the bubbles leading to protein removal from the bubble’s film [30].”
Comment: L372 - If authors do not have data, there is not need to include the information.
Response: The authors disagree with this statement. Treatment with protease resulted in no foaming, which suggested that proteins are required for the foaming capabilities observed in the faba bean precipitate. The authors do not see value in producing a table or figure demonstrating a negative or no-result, when we have included this statement as part of the discussion.
Comment: L373 - In which figure/table the authors observed no foaming was observed?
Response: See previous response to the last comment. No foaming was observed when the FBA-P was treated with either protease or a-amylase. The authors do not see value in producing a table or figure demonstrating a negative or non-result, when we have included this statement as part of the discussion.
Comment: L376 - alpha amylase; please correct typo
Response: Typo has been corrected.
Comment: L391 - What do you mean when you refer to ''isolation''? I was not able to see a single molecule isolated. The authors obtained different fractions with multiple composition.
Response: “isolated” was removed and L393 now reads “Faba bean process water was characterized to assess its foaming capacity and composition.”
Comment: That would be interesting is authors include extraction of chickpea and analyze the aquafaba as a control. How comparable is fava bean with chickpea?
Response: The faba bean process water demonstrated similar functional properties (e.g., foaming capacity and foaming stability) as chickpea wastewater (aquafaba).

Reviewer 2 Report
Dear authors,
After reading the manuscript , I realized that the manuscript showed in some parts the scientific rigour wanted, but in other parts I have missed it.The authors have presented critical evaluation only in some paragraphs.
The references are not exactly current, besides the objective and title could be more improved.
Thats why I have written some suggestions below in an attempt to improve the paper.
L.31- Avoid words that already are in your title
L.35- I suggest re-thinking the term "healthier" in the case of your paper. If faba beans replaced egg whites, that doesn't seem unhealthy to me. The approach was not well chosen, for a vegan it would be cultural perhaps.
L.46- author ??
L.71- Reading you title and your objective, in my humble opinion some things are missing. For example : Culinary X "to develop a process for isolating and fractionating process water from faba beans" "solution for its functional properties, such as foaming capacity and stability", "starch, monosaccharide, protein, and peptide content, and evaluated the molecular weight "
L.109- I missed authors in Material and Methods
L.115- Pore size ??
L.135- whose preliminary findings? yours ?
L.144- "Analysis of Starch Content" - Where are the results ? It is confusing.
L.152 and 161- Did you compare methods ? Be clearer
L.266- I don't see sustainability or green cuisine in the paper, I see chemical analysis.
L.285- Shouldn't you have compared it to egg whites to get a realistic control?
L.289- NA ou ND ??
L.355- I suggest changing red color in Figure 4 for another one.
L.384- I missed the discussion of some results, for example L.Dopa, which was even included in the conclusion.
L.398 and 401 - Was evaluating L. Dopa one of your objectives?
Accept the chemical analysis of your study in the title and objectives, the paper has merit in terms of originality and deep analysis, but it can't stay the way it is, you need to adjust the approach, which is neither sustainability nor culinary.
The authors could add more current references, at least for 2022.
Minor editing of English language required
Author Response
Reviewer #2:
Comment: The references are not exactly current, besides the objective and title could be more improved.
Response: The title has been changed and Lines 69-71 now reads “The focus of this study was to characterize the different fractions obtained from faba bean process water and to investigate its functional properties compared to sustainable aquafaba (i.e., chickpea wastewater).”
Comment: L.31- Avoid words that already are in your title
Response: Keywords have been changed to better reflect the manuscript. However, the authors think inclusion of “faba bean” and “aquafaba” should still be included as keywords.
Comment: L.35- I suggest re-thinking the term "healthier" in the case of your paper. If faba beans replaced egg whites, that doesn't seem unhealthy to me. The approach was not well chosen, for a vegan it would be cultural perhaps.
Response: L35 has been changed and now reads “This shift is largely due to increased consumer demand for more sustainable plant-based ingredient options.”
Comment: L.46- author ??
Response: The reference has been corrected.
Comment: L.71- Reading you title and your objective, in my humble opinion some things are missing. For example: Culinary X "to develop a process for isolating and fractionating process water from faba beans" "solution for its functional properties, such as foaming capacity and stability", "starch, monosaccharide, protein, and peptide content, and evaluated the molecular weight "
Response: The authors thank the reviewer for this comment and have added some clarification to Lines 69-71, which now reads “The focus of this study was to characterize the different fractions obtained from faba bean process water and to investigate its functional properties compared to sustainable aquafaba (i.e., chickpea wastewater).”
Comment: L.109- I missed authors in Material and Methods
Response: The cotyledons were separated from dehulled faba bean and is described in Section 2.2. Aqueous extraction of the cotyledons is then explained in Section 2.3.
Comment: L.115- Pore size ??
Response: Pore size has been included (pore size was 1 mm).
Comment: L.135- whose preliminary findings? yours ?
Response: The findings are based on our own preliminary findings. Lines 135-136 have been clarified and now reads “These methods were developed based on our previous preliminary findings (Reaney, University of Saskatchewan, 2022, personal communication)…”
Comment: L.144- "Analysis of Starch Content" - Where are the results ? It is confusing.
Response: The results are listed in Section 3.1. in the Results and Discussion.
Comment: L.152 and 161- Did you compare methods ? Be clearer
Response: GC-FID was used to confirm/qualify the monosaccharide content in the FBA-P fraction. Clarification was added to:
Lines 161-163 which now reads “Another aliquot of the lyophilized sample was separately prepared for gas chromatography equipped with a flame ionization detector (GC-FID), described below).”
Line 164 which now reads “Monosaccharide content was further qualified using GC-FID.”
Comment: L.266- I don't see sustainability or green cuisine in the paper, I see chemical analysis.
Response: We are unsure what the reviewer is commenting to on Line 266. If the reviewer is commenting on the title, the authors have decided to change the title to “Foaming with Starch: Exploring Faba Bean Aquafaba as a Green Alternative”.
Comment: L.285- Shouldn't you have compared it to egg whites to get a realistic control?
Response: Although the authors do agree that it would have been interesting to include egg whites, the purpose of this study was to investigate whether faba bean process water exhibited similar properties to chickpea process water (aquafaba).
Comment: L.289- NA or ND ??
Response: NA or Not Applicable. The Table has been updated.
Comment: L.355- I suggest changing red color in Figure 4 for another one.
Response: The red colour in Figure 4 has been changed to black.
Comment: L.384- I missed the discussion of some results, for example L. Dopa, which was even included in the conclusion.
Response: A short discussion of L. dopa has been included into the results and discussion. This was specifically added to Section 3.5 Lines 388-392 and now reads “The presence of levodopa (L-Dopa) is of interest, as this compound is a dopamine precursor which has been used in the treatment of Parkinson’s disease and hormonal imbalance [17, 33–38]. The synthesis of this compound can be influenced by soil nitrogen and phosphorus content during growth [39] and is also prone to degradation during cooking or in the presence of alkaline solutions [40–42]. Nonetheless, isolating and enriching this value-added compound, from faba bean process water, can provide producers new market entries.
Comment: L.398 and 401 - Was evaluating L. Dopa one of your objectives?
Response: No L-Dopa was not one of our objectives. However, we did observe the presence of this compound in the FBA-S, which could be of interest as an added-value product from faba bean process water.
Comment: Accept the chemical analysis of your study in the title and objectives, the paper has merit in terms of originality and deep analysis, but it can't stay the way it is, you need to adjust the approach, which is neither sustainability nor culinary.
Response: The title has been changed “Foaming with Starch: Exploring Faba Bean Aquafaba as a Green Alternative”.
Comment: The authors could add more current references, at least for 2022.
Response: More recent references have been included into the introduction.

Round 2
Reviewer 2 Report
After another evaluation of the manuscript, I realized a great improvement in the quality of the paper. The authors have accepted most of my requests.English is always useful to ask a native speaker for a final appreciation.
They added more authors to better substantiate Introduction, methodology and corrected tables and graphs.
After the corrections made by the authors, this final version of the paper is definitely much better.
Some more points to review:
L.152- "preliminary findings, (Reaney, University of Saskatchewan, 2022, personal communication) - Don't you have the publication with authors and year ?
L.289- For me, 3.1 appears as "Aquafaba production and properties" - Take a look, please. It is not clear yet.
Comment: L.144- "Analysis of Starch Content" - Where are the results ? It is confusing.
Response: The results are listed in Section 3.1. in the Results and Discussion.
Minor editing of English language required
Author Response
Foaming with Starch: Exploring Faba Bean Aquafaba as a Green Culinary Alternative
Thank you for your patience and recommendations for strengthening our manuscript (ID: foods-2586727). We have revised our manuscript according to Reviewer 2’s comments. We hope these changes improve the overall quality of this manuscript for publication. We have listed Reviewer 2’s comments and answered them in sequence.
Reviewer #2:
Comment: L.152- "preliminary findings, (Reaney, University of Saskatchewan, 2022, personal communication) - Don't you have the publication with authors and year?
Response: We have revised the manuscript by rewriting this sentence (Line 136) and placing as a rationale for measuring starch content (Lines 119, 120). The preliminary research led to this research.
Comment: L.289- For me, 3.1 appears as "Aquafaba production and properties" - Take a look, please. It is not clear yet.
Response: We agree that the section title is misleading. We have revised it to a more descriptive title. “1H-NMR evidence shows that functional aquafaba precipitates contain starch”. To avoid confusion, this subsection has been renamed to “3.1. Aquafaba production, functional properties, and starch content” (Line 269).
Comment: L.144- "Analysis of Starch Content" - Where are the results? It is confusing.
Response: The title was misleading as the spectra of an amylose standard were recorded and compared with the spectrum of FBA-P (Line 144). Both the FBA-P spectra and the a-amylose spectra of products dissolved in deuterated DMSO are provided in Section 3.1 of the Results and Discussion specifically Lines 293 to 300.
